# An Edible and Quick-Dissolving Film from Cassia Gum and Ethyl Cellulose with Improved Moisture Barrier for Packaging Dried Vegetables

**DOI:** 10.3390/polym14194035

**Published:** 2022-09-27

**Authors:** Tingting Li, Fansong Meng, Wenrui Chi, Shiyu Xu, Lijuan Wang

**Affiliations:** Key Laboratory of Bio-Based Materials Science and Technology of Ministry of Education, Northeast Forestry University, 26th Hexing Road, Xiangfang District, Harbin 150040, China

**Keywords:** edible films, quick-dissolving performance, improved barrier properties, packaging film for dried vegetables

## Abstract

A quick-dissolving edible film was made from cassia gum (CG) incorporated with ethyl cellulose (EC). Mechanical results show that addition of 5% EC based on CG gave rise to the highest tensile strength (TS) of the composite film. Scanning electron microscopy revealed that excess addition of EC slightly decreased the homogeneousness of films. Fourier transform infrared spectroscopy showed that the compatibility between CG and EC was good and the incorporation of EC changed the original interaction of molecules by forming hydrogen bonds with CG. Although film light transmittance decreased, it is transparent enough for packaging. The film water vapour barrier property improved dramatically by blending CG and EC, although they showed dissolution rates over 80% in boiling water after 5 min. The dried carrot cube packaged by CG-EC films showed lower mass growth rates in 53% RH. Therefore, the film presents a potential application in packaging of dried vegetables in convenience foods.

## 1. Introduction

Instant noodle is one of the most popular convenience foods all over the world. Its global consumption increased to 116.56 billion with a growth rate of 9.5% in 2020. It is known that the special flavour of cooked noodles depends on seasonings, such as dried vegetables and oil-rich sauces packaged separately in small bags. These packaging bags in most cases are made from petroleum-based materials including polyethylene, polypropylene, and polyvinyl chloride [1]. Consumers must take out the seasonings and tear up the small bags, as they are neither edible nor soluble in water; this does not meet the need of super convenience. Furthermore, the heavy uses of plastic bags is accelerating the accumulation of white rubbish, which is very difficult for microorganisms to degrade within a few decades [2]. Nowadays, researchers are trying to replace petroleum-derived plastics in food packaging using biobased films that are mainly made of polysaccharides [3,4] lipids [5], proteins [6], and composites. Among these, polysaccharides are the most promising materials thanks to their rich sources, edibility and natural regeneration. Moreover, most of them can quickly dissolve in water, meaning that people can cook polysaccharide-based bags packing seasonings together with noodles in water directly, avoiding the waste and pollution problem. Moreover, polysaccharides have been found to have excellent film-forming performances with potential in packaging [3]. However, most studies on polysaccharide-based films have focused on their applications in packaging meat, oil-rich food, and fresh fruits and vegetables [7,8], not in packaging dried vegetables. Rice starch–pectin films with green tea extract (GTE) have exhibited excellent antioxidation properties, and can be used as active food packaging. However, the incorporation of GTE lowers the tensile strength [9]. A film composed of yellow peach peel, sodium alginate, and glycerol with outstanding antioxidation property significantly decreased peroxide value (POV) to delay oil oxidation [10]. However, its poor mechanical property cannot meet the requirements of dried vegetables packaging. These films exhibit appropriate tensile strength, flexibility, transparency, gas barrier properties, and even antibacterial [11] and antioxidant activity. However, they are not suitable for packaging dried vegetables.

Because dried vegetables easily absorb moisture, the packaging should protect them from water and external forces. However, high water vapor permeability (WVP) and poor mechanical properties present an ongoing challenge with respect to polysaccharide-based films [12]. To solve this problem, reinforcers can be introduced into the pure film. The addition of 4% beeswax decreased WVP and resulted in a significant decrease in tensile strength, elongation at break, and Young’s modulus of cordia gum films [13]. This may be due to the phase separation caused by the polar difference between polysaccharides and lipids, making the film less continuous and homogeneous. Carboxymethyl chitosan/locust bean gum films incorporated with a *Melissa officinalis* L. essential oil nanoemulsion presented a lower WVP and tensile strength (TS), which may be due to the aforementioned weak compatibility [14]. Films based on plantago major seed gum containing increasing contents of olive oil showed an enhancement in the water barrier property but a loss in TS [15]. A similar phenomenon has been found in starch-based films containing black cumin oil, oregano oil, and clove essential oils [16,17]. Therefore, we propose a path to simultaneously obtaining a higher WVP barrier and tensile strength by replacing oils with hydrophobic polymers.

Cassia gum (CG) is a natural and edible polysaccharide from the leguminous plant *Cassia tora* (*Cassia obtustifolia*) [18]. It presents good film-forming properties thanks to plasticizers such as glycerin and sorbitol [19]. As such, it is a candidate for edible packaging films. CG films have exhibited the good mechanical properties and seal strength after incorporation with carboxylated cellulose nanocrystal whiskers [20]. It can be used as a carrier for natural antioxidants such as rutin or quercetin for the preservation of lipids [21,22]. However, poor mechanical properties and WVP barrier limit its wider applications.

Ethyl cellulose (EC) is a derivative of cellulose. Compared to most cellulose derivatives, it shows particular hydrophobicity, and can dissolve in organic solvents but not water when its degree of substitution is between 2.0 and 2.4. It can form a film in which the free –OH groups in molecular chains can form hydrogen bonds, leading to a dense reticular structure which favors increased hydrophobicity and mechanical properties in EC films [23]. As an additive, EC is more miscible and stable thanks to its high molecular weight and excellent thermal stability compared with beeswax or essential oils [24]. Therefore, it is more promising for increasing the moisture barrier of films. Higher water resistance has obtained by blending EC in konjac glucomannan-based film [25]. To the best of our knowledge, CG films containing EC for improved properties and potential in packaging dried vegetables have not been reported yet.

The aim of this study was to develop CG films with higher TS and moisture resistance by incorporating EC. The films were characterized by SEM and FTIR spectroscopy. Then, they were investigated in terms of their mechanical and barrier properties as well as their solubility in water. Finally, a simulative application was carried out in order to study the effects of films on protecting dried vegetables.

## 2. Materials and Methods

### 2.1. Materials

Cassia gum was bought from Laweiya Biological Co., Ltd. (Xian, China). EC (white powder, a nonionic cellulose ether insoluble in water and soluble in organic solvents, 2.26 < DS < 2.36, ethoxy group content (44–51%), viscosity (40–52 Pa•s), was purchased from Heda Co., Ltd. (Zibo, China). Glycerol was provided by Damao Chemical Co., Ltd. (Jinan, China). Dried carrot cube was purchased from a local market in Harbin.

### 2.2. Preparation of CG-EC Films

CG powder 6.0 g and glycerol 2.7 g were mixed in 670 mL of distilled water and stirred at a speed of 450 rpm for 1 h. Then, 25 mL of 75% ethanol was used to dissolve EC powders (1%, 2%, 3%, 4%, 5%, or 6% based on CG mass), which was added to the CG solution. After stirring, the solution was poured into the plate and dried at 60 °C for 36 h. Finally, the film was peeled off and kept in a sealing bag. Films were named as CG-x%EC according to the content of EC, where x is 0, 1, 2, 3, 4, 5, and 6, respectively.

### 2.3. Characterization

#### 2.3.1. Fourier Transform Infrared Spectroscopy (FTIR)

FTIR spectra of the films and raw materials were obtained using an infrared analyzer (Thermo Fisher, Waltham, MA, USA) from 4000 to 600 cm^−1^ at a resolution of 4 cm^−1^.

#### 2.3.2. Scanning Electron Microscopy (SEM)

The micromorphology of films was obtained by a Quanta 200 SEM (Philips-FEI Co., Eindhoven, The Netherlands) using sprayed gold under a high vacuum.

### 2.4. Properties

#### 2.4.1. Thickness and Mechanical Properties

The thickness of films was obtained by a digital micrometer (ID-C112XBS, Tokyo, Japan). Each film was measured at fifteen different points to obtain the average thickness. According to ISO527-3, the TS and elongation at break (EB) were examined using a tensile testing machine (XLW-PC, Labthink, Jinan, China). Films were cut into specimens of 15.0 mm × 80.0 mm and conditioned at 53% RH for 12 h. The stretching speed and grip separation distance were set as 300 mm/min and 50 mm, respectively.

#### 2.4.2. Water Vapor Permeability (WVP)

The WVP of films was measured according to the method in [26] with a few modifications. Circular films with a radius of 3.0 cm were sealed on the mouth of a weight bottle containing 23.0 g of CaCl_2_ with hot melt adhesive. The weight bottles were weighed originally and put into a desiccator with 53% RH from saturated Mg(NO)_3_ solution, then weighed in order after 0.5, 1, 1.5, 2, 3, 4, 5, 6, 8 and 10 h, respectively. The WVP values were obtained using Equation (1):WVP = (Δm × d)/(A × Δt × Δp)(1)
where Δm (g) is the mass difference between the test sample and the initial sample and A (m^2^) and d (m) are the area and thickness of the samples, respectively.

#### 2.4.3. Light Transmittance, Transparency, and Haze

The transmittance and haze of the films were measured by a UV spectrophotometer (UV-2600, Shimadzu, Kyoto, Japan) and a haze meter (CS-700), respectively. The transparency of the films was calculated with Equation (2):Transparency = T_600_/d(2)
where T_600_ (%) and d (μm) are the transmittance at 600 nm and the thickness, respectively.

#### 2.4.4. Thermogravimetric Analysis

The thermal properties of the films were measured using a TA Instruments TGA Q500 (TA Instruments, New Castle, DE, USA) in a temperature range of 10 to 600 °C with a speed of 10 °C/min.

#### 2.4.5. Dissolution Rate (DR) of Films

Films were dried at 103 °C and weighed as W_1_. Then, they were placed into the boiling water for 1, 2, 3, 4, 5, and 6 min. The insoluble residues were separated and dried at 103 °C to a constant weight as W_2_. The DR was calculated using Equation (3):Dissolution rate = (W_1_ − W_2_)/W_1_ × 100%(3)
where W_1_ (g) and W_2_ (g) are the masses of the initial sample and the residue, respectively.

### 2.5. Application

Films were cut into rectangles of 6.5 cm × 6.5 cm. Two pieces were sealed to form a bag using a sealing machine for packaging a dried carrot cube with an original weight of W1. The bags were put into an environment of 53% RH and the dried carrot cubes inside them were taken out and weighed as Wn after 12, 24, 72, and 120 h. The same mass of dried carrot cube was directly exposed to 53% RH as a control. The mass growth rate and inhibition rate were calculated using Equations (4) and (5), respectively:Mass growth rate (MGR) = (W_n_ − W_1_)/W_1_ × 100%(4)
Inhibition rate = (MGR_control_ − MGR_n_)/MGR_control_ × 100%(5)
where W_1_ (g) and W_n_ (g) refer to the mass of the initial sample and test sample after 12, 24, 72, and 120 h, respectively, and MGR_control_ (%) and MGR_n_ (%) refer to the mass growth rate of the dried carrot cube without packing and when packed using different films, respectively.

### 2.6. Statistical Treatment

Data are expressed as mean values ± standard deviation. SPSS was used to analyze the differences among the data and confirm the statistical significance if *p* < 0.05.

## 3. Results and Discussion

### 3.1. FTIR Analysis

Figure 1 shows the FTIR spectra of pure EC film, pure CG film, and CG/EC films. In the spectrum of EC film, the peak at 3462 cm^−1^ was designated to the –OH stretching vibration peak. The stretching vibration peak of –C_2_H_5_ was observed at 2874 cm^−1^ and 2975 cm^−1^. Another sharp peak at 1375 cm^−1^ was assigned to the stretching vibration of –CH_2_–. The broad band around 3273 cm^−1^ in the spectrum of pure CG film was due to hydrogen bonds between CG molecules. The peak at 2874 cm^−1^ belonged to C–H stretching. The bending vibration peak of C–H was observed at 1320 cm^−1^. Peaks at 1591 and 1412 cm^−1^ were due to vibration coupling of –COO−, which may be derived from glucuronic acid in the CG molecular chain [27]. Compared with pure CG film, the band of –OH in the spectrum of CG-0%EC shifted from 3273 cm^−1^ to 3260 cm^−1^ because of the hydrogen bonds between CG and glycerin. After 5% EC was added, the band of –OH shifted sequentially from 3260 cm^−1^ to 3249 cm^−1^, meaning that more hydrogen bonds formed. Except for a slight shift of the –OH bands, no other change was found in the spectra, indicating that the compatibility between CG and EC is good.

### 3.2. Micromorphology Analysis

The micrographs of surfaces and cross-sections of CG-0%EC, CG-1%EC, CG-3%EC, CG-5%EC, and CG-6%EC are shown in Figure 2. A smooth surface was observed in CG-0%EC, suggesting that CG had good film-forming performance. There was no significant difference except for a slight decrease in uniformity as the EC content increased from 1% to 5%. However, when it increased to 6%, uneven particles appeared on the surface due to the high concentration of ethyl cellulose. As can be seen, while CG-0%EC showed a homogeneous cross-section, a few veins were distributed evenly as the EC content increased from 1% to 5%. In addition, particles and cracks appeared in the CG-6%EC cross-section because excess content of EC in the film disrupted the network, which is in agreement with the microstructure of the surfaces.

### 3.3. WVP Analysis

The WVP results are shown in Figure 3. As the content of EC increased from 0% to 6%, the WVP showed an obvious decreasing trend from 2.28 × 10^−10^ to 1.40 × 10^−10^ g m^−1^ s^−1^ Pa^−1^, which was lower than other results for Plantago major seed gum with maize oil (1.1 × 10^−9^ g m^−1^ s^−1^ Pa^−1^) [15], *Lepidium perfoliatum* seed gum with stearic acid (1.68 × 10^−10^ g m^−1^ s^−1^ Pa^−1^) [28], and starch film with yerba mate extract (4.6 × 10^−10^ g m^−1^ s^−1^ Pa^−1^) [29]. This phenomenon was due to the hydrophobicity of EC. As the EC content increased, the hydrophilic/hydrophobic ratio of the film-forming matrix decreased, resulting in enhancement of the water vapor barrier property, similar to the study in [30]. Furthermore, the saturation property of hydrogen bonds between CG and EC made it difficult for water vapor to pass through the film.

### 3.4. Thickness and Mechanical Properties Analysis

The thickness is shown in Table 1, different superscripts in each column are significantly different (*p* < 0.05). EC led to an increasing trend of thickness when its content increased from 0% to 6%, which was caused by the higher solid content introduced into the matrix.

The effects of EC on the TS and EB are shown in Table 1 and Figure 4. As the content of EC increased from 2% to 5%, the TS increased from 31.75 ± 0.75 MPa to 38.77 ± 0.57 MPa. A possible mechanism diagram is shown in Figure 5. In the film-forming process, the hydrophobicity of EC may facilitate EC chains to form a stable annular shape with the evaporation of ethanol. In contrast, CG is hydrophilic because of many –OH groups, which allow CG chains to move irregularly and become tangled with EC chains. Finally, a denser interpenetrating network was formed to promote the TS. As the content of EC increased to 6%, the TS decreased to 33.29 ± 0.95 MPa. Part of the EC precipitated because the concentration of EC increased in line with the evaporation of ethanol, which weakened the uniformity of the film. In summary, an optimum content of EC facilitated the formation of a strong network and increased TS. However, excess EC decreased their enhancement effect, which was consistent with SEM results. A similar phenomenon was found in κ-carrageenan films incorporated with pomegranate flesh and peel extracts [31]. On the other hand, the increase in TS was coupled with a decrease in EB. As the EC content increased from 0% to 6%, EB decreased from 27.80 ± 0.71% to 10.90 ± 1.35%. This result was due to deposited EC, which reduced the movement of the molecule chains. Another reason is that the plasticizing effect from water vapor was weakened because of the decrease in WVP.

### 3.5. Light Transmittance and Haze Analysis

As shown in Figure 6a, transmittance decreased significantly as EC content increased. However, in the visible region it remained higher than 50% even when EC content increased up to 6%. The haze and transparency indexes are shown in Figure 6b. As the EC content increased from 0% to 6%, haze increased from 0.9% to 22.5% and the transparency index decreased from 1.54 to 1.02. This could be due to the aggregation of EC, which can intensify light scattering, thereby increasing haze.

### 3.6. DR Analysis

The DR values of films are shown in Figure 7a. As the boiling time increased from 1 min to 6 min, the DR of CG-0%EC and CG-5%EC increased from 58.74 ± 3.52% and 48.75 ± 2.5% to 93.84 ± 1.29% and 86.40 ± 1.04%, respectively. Their high DR values were due to CG being soluble in water. CG-5%EC showed a lower DR than CG-0%EC because the incorporation of EC decreased the hydrophilicity of the film. Figure 7b shows that higher TS of CG-5%EC helped the film to maintain a relatively complete shape instead of breaking into fragments in the boiling water.

Figure 7c shows photos of packaging bags made from CG-0%EC and CG-5%EC after soaked in warm water (60 °C) and boiling water under slight shaking. The bag from CG-0%EC incurred obvious deformations at 15 s and broke at 30 s. After heating for 1 min, part of the bag dissolved, although residue could be observed. As the boiling was prolonged to 2 min, the main body of the bag dissolved. Compared to CG-0%EC, CG-5%EC presented better performance against water. At 15 s, a slight wrinkle appeared on the bag; at 30 and 60 s, a severe wrinkle was observed. Even when the soaking time increased to 60 s, no breakage occurred. Finally, it dissolved after being heated for 2 min. Although the bag made from CG-5%EC showed effective ability against water, it was able to dissolve in the boiling water in a short time. This phenomenon was due to the high temperature accelerating the motion of water molecules, allowing them to break the hydrogen bonds. This high solubility in boiling water could allow it to be used for packaging seasonings in convenience foods.

### 3.7. Thermogravimetric Analysis

The thermal stability of the raw materials and films in the temperature range of 10 to 600 °C is shown in Figure 8. Glycerin, CG, and EC showed single peaks at 220.3 °C, 279.4 °C, and 338.3 °C, respectively. As for the films, they showed two main processes of weight loss. The first was observed around 115 °C, which was ascribed to the evaporation of water inside the films. The second occurred in the range of 217 to 317 °C due to the thermal decomposition of the main film-forming substances. CG-0%EC showed a peak at 256.0 °C, which was lower than that of CG powder. This phenomenon was caused by the decrease in the degree of crystallinity during the film-forming process. Additionally, the decomposition peak of CG-6%EC was observed at 255.5 °C, which means that the incorporation of EC had no obvious influence on the film decomposition temperature.

### 3.8. Application of Films in Packaging Dried Carrot Cube

The inhibition rates of the films are shown in Figure 9. After 12 h, inhibition rates increased from 28.98% to 43.34% as EC content increased from 0% to 5%. In the same period, CG-5%EC presented the highest inhibition rate, which is consistent with the WVP analysis. Another reason for this was that the increased thickness extended the distance the water vapor needed to pass through. As the time was extended to 120 h, the inhibition rate presented a decreasing trend, as the water vapour barrier property was weakened by the invasion of more and more water molecules. The result show that packaging made from CG-5%EC may extend the shelf life of dried vegetables by protecting them from water vapor.

## 4. Conclusions

A series of quick-dissolving moisture barrier CG/EC films for vegetable packaging were successfully prepared. The incorporation of EC facilitated a strong network by forming hydrogen bonds with CG, which was confirmed by FTIR. The CG-5%EC film showed the best TS at 38.77 ± 0.57 MPa. Veins were observed on the cross-section as EC content increased from 0% to 5%. The addition of EC endowed CG film with decreased transmittance and increased haze. Due to the hydrophobicity of EC, the WVP of films showed a decreasing trend to 1.40 × 10^−10^ g m^−1^ s^−1^ Pa^−1^. When CG-5%EC was applied to dried carrot cube packaging, the inhibition rate of water reached to 43.34% after 12 h. Although the packet made from CG-5%EC presented a lower dissolution rate than CG-0%EC, it dissolved in boiling water within 2 min. The results indicate that the edible CG-5%EC film has potential in dried vegetable packaging for convenience foods while achieving the goal of environmental protection.

## Figures and Tables

**Figure 1 polymers-14-04035-f001:**
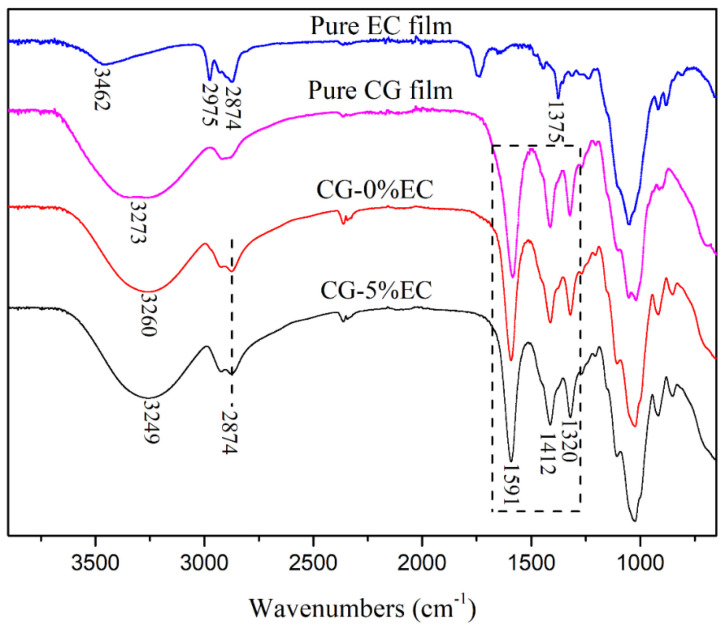
FTIR spectra of pure CG film, pure EC film and CG/EC films.

**Figure 2 polymers-14-04035-f002:**
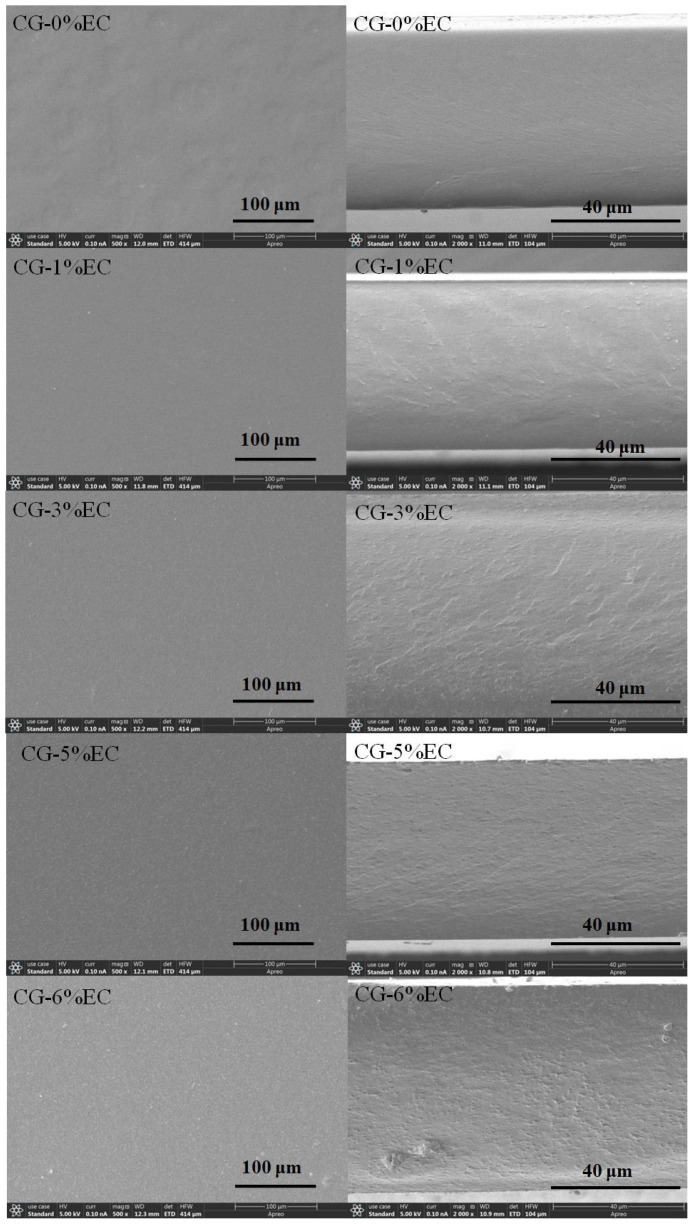
SEM micrographs of surface (**Left**) and cross-section (**Right**) of CG/EC films.

**Figure 3 polymers-14-04035-f003:**
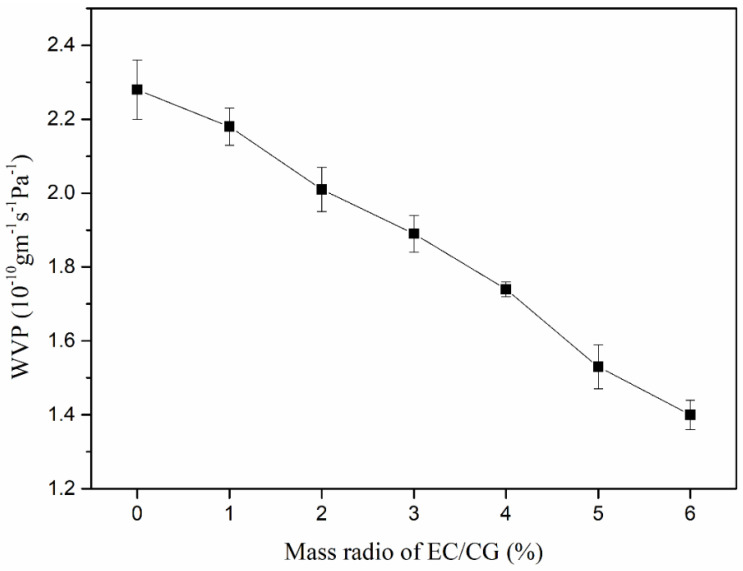
The relationship between WVP and the ratio of CG/EC.

**Figure 4 polymers-14-04035-f004:**
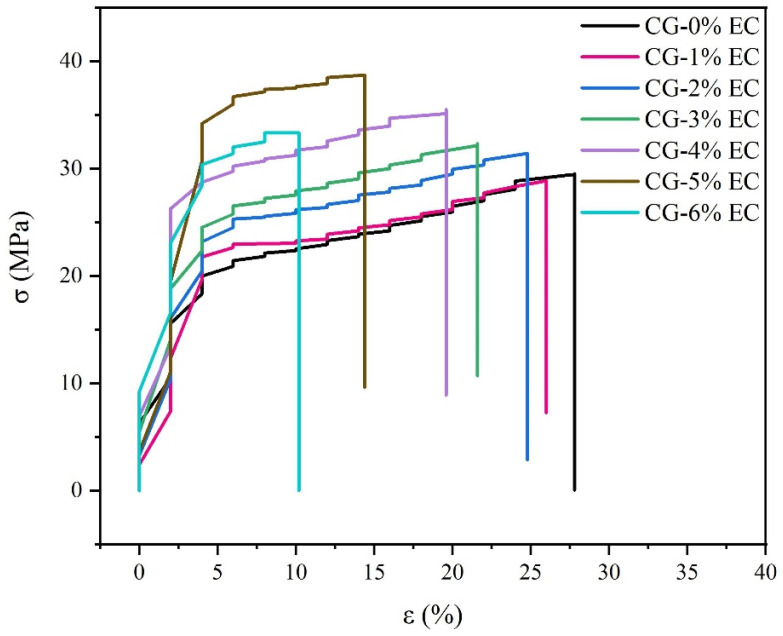
The stress–strain curves of films incorporated with different amounts of EC.

**Figure 5 polymers-14-04035-f005:**
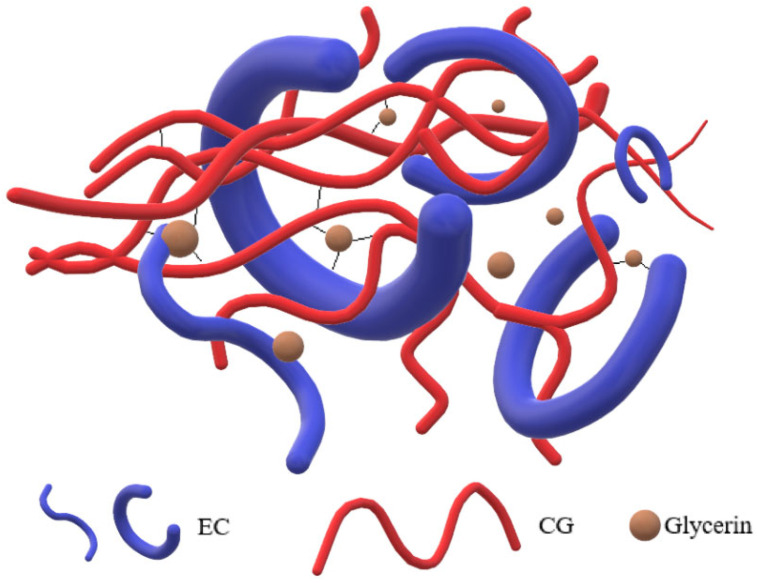
Mechanism diagram of intermolecular action.

**Figure 6 polymers-14-04035-f006:**
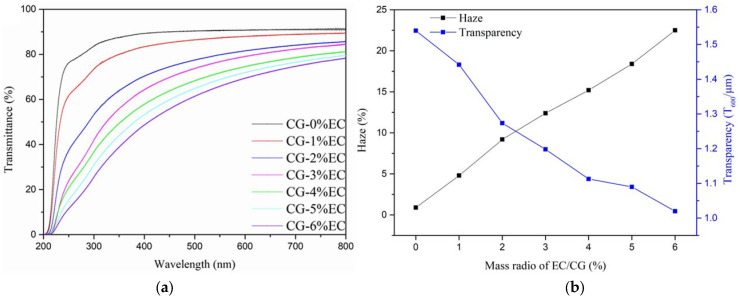
The light transmittance (**a**) and haze and transparency (**b**) of the films.

**Figure 7 polymers-14-04035-f007:**
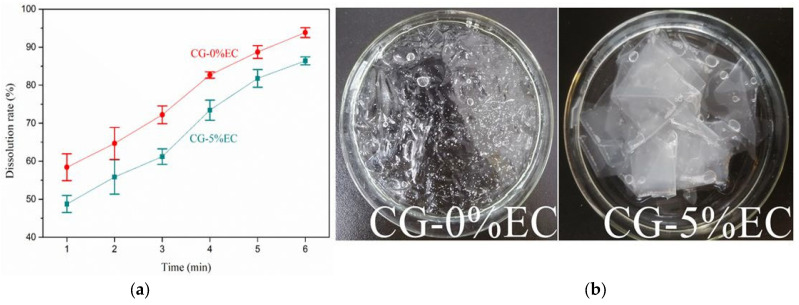
DR (**a**) and soaking photos (**b**,**c**) of films.

**Figure 8 polymers-14-04035-f008:**
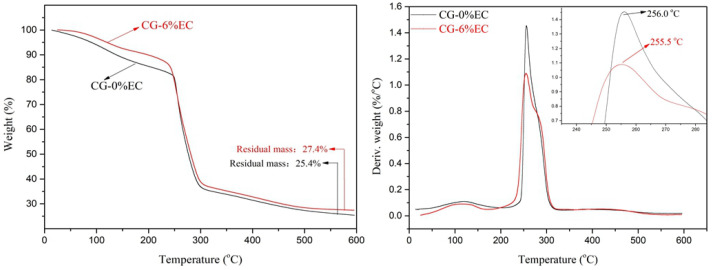
TG and DTG Analysis of raw materials and films.

**Figure 9 polymers-14-04035-f009:**
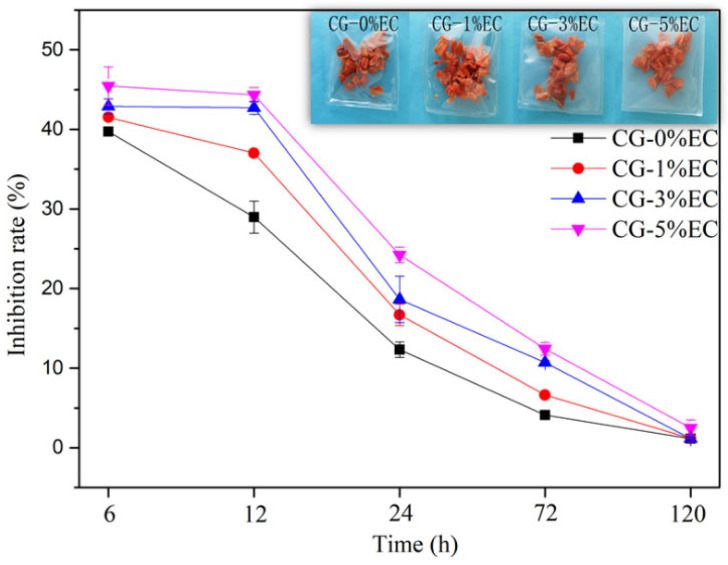
The effects of time on inhibition rates of water absorption of carrot cubes packaged by different films.

**Table 1 polymers-14-04035-t001:** Thickness, TS, and EB of films incorporated with different amounts of EC.

Sample	Thickness (μm)	TS (MPa)	EB (%)
CG-0%EC	59.75 ± 2.34 ^a^	29.30 ± 0.21 ^a^	27.80 ± 0.71 ^f^
CG-1%EC	61.13 ± 2.31 ^ab^	28.58 ± 0.86 ^a^	25.75 ± 0.73 ^e^
CG-2%EC	64.70 ± 2.22 ^c^	31.75 ± 0.75 ^b^	25.25 ± 0.90 ^e^
CG-3%EC	63.29 ± 1.46 ^bc^	32.85 ± 0.31 ^bc^	21.30 ± 0.77 ^d^
CG-4%EC	67.33 ± 2.66 ^d^	35.84 ± 1.18 ^d^	19.35 ± 1.17 ^c^
CG-5%EC	66.19 ± 2.45 ^cd^	38.77 ± 0.57 ^e^	14.73 ± 0.97 ^b^
CG-6%EC	68.75 ± 2.23 ^d^	33.29 ± 0.95 ^c^	10.90 ± 1.35 ^a^

## Data Availability

The data presented in this study are available on request from the corresponding author.

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
