# Peer review of "An Edible and Quick-Dissolving Film from Cassia Gum and Ethyl Cellulose with Improved Moisture Barrier for Packaging Dried Vegetables"

_polymers, 2022, doi:10.3390/polym14194035_

Round 1

Reviewer 1 Report

Li et al studied a quick dissolving and edible film from cassia gum (CG) and ethyl cellulose (EC). This film exhibits an improved moisture barrier for potential food packaging applications. The results show that the incorporation of 5% EC results in higher mechanical properties like the tensile strength of the edible film. These improvements are promising for their use as packaging applications. The paper and its subject are interesting. however, it can be further improved and can be considered after minor revision. A few points are –

[1] The introduction is well written. However, please provide 4-5 references from MDPI on the subject of research and highlight the advancement and novelty of the present work over existing literature in MDPI.

[2] In the material section, the details of EC are missing. Please also provide the physical and chemical properties of EC in this section.

[3] In section 2.4.1, what ISO standards are followed?

[4] In Figure 2, the resolution scale is not visible. Please insert the scale manually.

[5] Table 1 is not sufficient. Please provide stress-strain curves of all samples and describe their mechanical behavior briefly.

[6] Please justify why the authors studied TGA. How they are correlated with packaging applications?

[7] The conclusion section is poorly written. Please brief the outcome of the present work more precisely. Please also insight importance of this work in packaging applications. Which sample is best and why?

Good Luck with the revisions!

Reviewer 2 Report

Dear Authors

Your article is interesting and well organized. The topic is described in a clear and concise way. There are some suggestions that I recommend in order to improve the manuscript (please see the attached version of your manuscript).

The Introduction part provides sufficient data on the topic with relevant references. The experimental part gives all details needed. Results consequently come and are based on the experiments done in this study work. All data presented in this section (tabular or graphs) are explained and achieve a visual effect.

The conclusion section is well written and provides an adequate summary of the work.
